# Reinforcing the Immunocompromised Host Defense against Fungi: Progress beyond the Current State of the Art

**DOI:** 10.3390/jof7060451

**Published:** 2021-06-06

**Authors:** Georgios Karavalakis, Evangelia Yannaki, Anastasia Papadopoulou

**Affiliations:** 1Hematology Department-Hematopoietic Cell Transplantation Unit, Gene and Cell Therapy Center, “George Papanikolaou” Hospital, 57010 Thessaloniki, Greece; giorgos.karavalakis@gmail.com (G.K.); eyannaki@uw.edu (E.Y.); 2Department of Medicine, University of Washington, Seattle, WA 98195, USA

**Keywords:** fungal infections, fungus-specific T cells, T cell immunotherapy

## Abstract

Despite the availability of a variety of antifungal drugs, opportunistic fungal infections still remain life-threatening for immunocompromised patients, such as those undergoing allogeneic hematopoietic cell transplantation or solid organ transplantation. Suboptimal efficacy, toxicity, development of resistant variants and recurrent episodes are limitations associated with current antifungal drug therapy. Adjunctive immunotherapies reinforcing the host defense against fungi and aiding in clearance of opportunistic pathogens are continuously gaining ground in this battle. Here, we review alternative approaches for the management of fungal infections going beyond the state of the art and placing an emphasis on fungus-specific T cell immunotherapy. Harnessing the power of T cells in the form of adoptive immunotherapy represents the strenuous protagonist of the current immunotherapeutic approaches towards combating invasive fungal infections. The progress that has been made over the last years in this field and remaining challenges as well, will be discussed.

## 1. Introduction

Fungal infections, either superficial or invasive, may affect both the immunocompetent and immunocompromised individuals. For vulnerable patients with underlying disease, invasive fungal infections (IFIs) represent a major cause of morbidity and mortality worldwide, resulting in nearly 1.5 to 2 million deaths per year [1,2]. Neutropenia, allogeneic hematopoietic cell transplantation (allo-HCT), solid organ transplantation (SOT), acquired or inherited immunodeficiencies and prolonged corticosteroid administration are the major predisposing factors for IFI [3], most frequently caused by *Candida*, *Aspergillus*, or *Cryptococcus* genera [4]. Despite the recent significant advances in antifungal pharmacotherapy, mortality rates associated with IFIs remain high, especially in immunocompromised patients, mainly due to delayed recognition and diagnosis of infection. Importantly, specific restrictions apply to antifungal drugs, associated with bioavailability to target tissues, toxicity, treatment failure, route of administration, activity spectrum, drug interactions and the most alarming, drug resistance [2]. Therefore, the development of effective and minimally toxic, alternative antifungal therapies overcoming the well-recognized limitations of conventional drug treatment, is a major unmet need.

In this review, we discuss therapeutic approaches beyond what is currently thought as the gold standard for the management of fungal infections (Figure 1), focusing on the progress and challenges of T cell immunotherapy against invasive fungal diseases.

## 2. Treatment of Invasive Fungal Infections: The Unmet Clinical Need

The number of patients at risk for invasive fungal infections is continuously increasing as a result of the emerging medical innovations and the use of immunomodulatory agents. In contrast, the rate of development of new antifungal drugs is inversely correlated to clinical demands; in fact, only one new class of antifungal drugs has been developed in the last 30 years (echinocandins). A fundamental challenge in anti-fungal drug development is the evolutionary relationship with conserved important biochemical and biological features, between humans and fungi and because of this, toxicity against yeast will be also directed to humans. Reasonably, the three classes of antifungal drugs (azoles, echinocandins and polyenes) target structures unique to fungi.

In addition, antifungal drug resistance is more and more frequently now recognized and multi-drug resistant fungi, such as *Candida auris*, have risen spontaneously over the globe. *Candida*
*auris* is a recently recognized yeast with high transmissibility, multi-drug resistance and adverse outcomes, causing invasive infections and outbreaks in health care facilities and intensive care units [5].

Given the above, the development of novel, alternative antifungal approaches that will overcome current limitations and meet therapeutic needs, becomes imperative in patients with compromised immunity.

## 3. Strategies to Reinforce the Host Defense against Fungi

Antifungal drugs often fail to eradicate the infection, mainly in immunocompromised individuals. Immunomodulators, including cytokines, monoclonal antibodies and recently check-point inhibitors, enhance the impaired host defense and are used as an adjunctive immune therapy for invasive fungal diseases.

### 3.1. Recombinant Cytokines

Cytokines are key players in controlling the homeostasis of the immune system and commercially available recombinant cytokines have exerted a potential adjunctive role to conventional antifungal therapy.

#### 3.1.1. Colony-Stimulating Factors (CSFs)

Neutrophil defects or chemotherapy-induced neutropenia are the main risk factors for the development of fungal infections. CSFs are a host-directed and non-specific therapy stimulating the production, maturation and activation of neutrophils, thus enhancing the host defense against a variety of pathogens. Three types of recombinant human CSFs are available: Granulocyte CSF (G-CSF), granulocyte macrophage CSF (GM-CSF), and macrophage CSF (M-CSF). Among CFCs, G-CSF is the most widely used and well-tolerated CFC, used adjunctly to restore neutrophil counts after myelosuppressive chemotherapy. G-CSF is a European Medicines Agency (EMA)- and United States Food and Drug Administration (FDA)-approved growth factor, stimulating the growth and differentiation of hematopoietic progenitor cells to neutrophils and enhancing phagocytosis and neutrophils’ antifungal activity in vitro and in experimental mouse models [6,7,8,9]. In clinical settings, G-CSF when used alongside with antifungal agents as an adjunctive treatment, resulted in faster neutrophil recovery and decreased risk of fungal infections in neutropenic individuals following intensive chemotherapy or bone marrow transplantation for hematological malignancies [10,11]. In the setting of immunodeficiencies, G-CSF alone or in combination with antifungal agents was highly effective against *Candida* infection and relapsing *Candida* meningoencephalitis in two patients with CARD9 immunodeficiency [12,13]. By a meta-analysis of 14 randomized controlled trials addressing the role of CSFs in chemotherapy-induced febrile neutropenia involving 1,553 participants, Mhaskar et al., reported that the use of CSFs (G-CSF or GM-CSF), resulted in faster neutrophil and fever recovery in those patients, although it did not improve overall mortality, including fungus–related mortality [14]. Prophylactic treatment of 206 allogenic HCT recipients with G-CSF, GM-CSF, or both in a phase IV randomized clinical trial comparing the effect of each of the CFCs or their combination, although showed no difference in the invasive fungal disease incidence, it resulted in significantly lower fungus-related mortality in the GM-CSF and G-CSF+GM-CSF groups as compared to G-CSF-alone group (1.47%, 1.45%, and 11.59%, respectively, *p* = 0.016) [15].

#### 3.1.2. Interferon-Gamma (IFN-γ)

IFN-γ, which is produced predominantly by NK cells and T lymphocytes, promotes the production of reactive oxygen species and reactive nitrogen intermediates and enhances phagocytosis by activating macrophages [16,17]. There are plenty of case reports and case series demonstrating that the adjunctive use of IFN-γ to boost antifungal immunity can provide therapeutic benefit against invasive mycoses [18], especially in patients with congenital or acquired immunodeficiencies, including Signal Transducer and Activator of Transcription 3 (STAT3)-deficiency or HIV infection [19,20], in whom the impaired immune responses predispose to fungal infections [21]. Notwithstanding the promising results from case series and two phase II trials with the use of IFN-g treatment, either alone or in combination with standard antifungal treatment [19,22], larger randomized clinical trials are needed to further validate its beneficial effect.

### 3.2. Antibodies

Monoclonal antibodies targeting fungal components, such as cell wall glycoproteins, and other surface molecules, such as heat shock proteins (HSPs), could offer passive protection against fungi. Anti-HSP90 (efungumab) and anti-β-glucan antibody (2G8 or 18B7) are two examples of monoclonal antibodies that have been evaluated as immunotherapy of fungal infections in clinical trials.

Efungumab is a single-chain variable fragment of a human monoclonal antibody that “grabs” onto fungal HSP90. In a randomized, blinded, multicenter trial, the use of lipid-associated amphotericin B in combination with Efungumab significantly improved the clinical outcome for patients with invasive candidiasis when compared with the amphotericin B monotherapy cohort [23]. Nevertheless, in November 2006, Europe’s Committee for Medicinal Products for Human Use (CHMP) refused marketing authorization of Efungumab, raising major concerns on the product quality and safety [24]. Those concerns included the lack of a suitable method to control aggregation in the drug substance and the high levels of *E. coli* host cell protein in the final product that could potentially be immunogenic in vivo. Additional safety concerns included the induction of cytokine release syndrome (CRS) and hypertension, which in March 2007 following re-examination, were removed by CHMP, considered manageable in clinical practice [24].

The monoclonal anti-β-glucan antibody 2G8 has shown protection against *Candida albicans*, *Aspergillus fumigatus* and *Cryptococcus neoformans* in vitro and in experimental mouse models [25,26]. Interestingly, the administration of 2G8 has proven safe in human immunodeficiency virus-infected patients with cryptococcal meningitis, while at high doses, it transiently reduced serum antigen levels (glucuronoxylomannan) [27]. Further clinical development of this monoclonal antibody was interrupted due to a lack of industrial support [28].

An alternative antibody-mediated protection against fungi is radioimmunotherapy, which is a combination of radiation and immunotherapy. Here, antifungal antibodies are linked to radioisotopes to specifically deliver cytocidal radiation to target sites [29], thus radioimmunotherapy offers several advantages over the standard antifungal therapy, including (i) cytocidal function, (ii) invulnerability to drug resistance mechanisms, (iii) its independence of the immune status of a host and of drug-drug interactions, and (iv) its potential to provide benefit by a single dose or a limited number of doses for the treatment of fungal diseases [29,30]; taken together, it seems to have the potential to overcome the limitations of currently available antifungal agents.

### 3.3. Checkpoint Inhibitors

Immune checkpoint inhibitors (ICIs) are novel modulators of immune homeostasis, targeting inhibitory receptors [e.g., programmed cell death protein 1 (PD-1), cytotoxic T lymphocyte antigen 4 (CTLA-4)] and ligands (PD-L1) expressed on T lymphocytes, antigen-presenting cells and tumor cells and aiming to stimulate immune cells to elicit an anti-tumor response. Despite being a paramount achievement in cancer treatment, ICIs claim also a place in antifungal treatment by reversing the hyporesponsiveness of innate and adaptive immunity during IFI [31]. In particular, the blockade of the negative co-stimulatory molecules, PD-1 and CTLA-4, improved survival in mouse models of primary and secondary fungal sepsis [32]. In a clinical setting, the administration of IFN-γ in combination with anti-PD-1 antibody resulted in an impressive clinical response in a patient with life-threatening mucormycosis, unresponsive to conventional therapy [33]. Targeting inhibitory immune checkpoints appears thus promising for the treatment of fungal infections in an era of increasing antifungal resistance.

### 3.4. Transfer of Innate Immune Cells to Control Fungal Infections

Neutropenia is a well-established risk factor for IFIs, especially aspergillosis and mucormycosis. Consequently, several studies focused on enhancing innate immunity by transfusion of granulocytes, NK cells or dendritic cells.

#### 3.4.1. Granulocyte Transfusion (GTX)

Neutrophils belong to the first line of defense and are necessary for direct fungal killing, while the risk of developing IFIs is inversely related to the number of circulating neutrophils. Theoretically, GTX, which includes transfusions of high numbers of neutrophils from a healthy ABO matched donor to a neutropenic patient who is at high risk of developing IFI, would increase a patient’s neutrophils and enhance the host’s defense against fungi, however, there is no solid evidence on GTX efficacy as a treatment of life-threatening fungal infections. By summarizing the existing evidence on GTX in IFIs from individual case reports, case series, matched cohort studies and clinical trials, West et al. concluded that while there is limited evidence that GTX may reduce the incidence of fungaemia, it does not prevent mortality [34]. Given the GTX-associated risks for the recipients, such as febrile transfusion reactions, pulmonary complications, platelet and leucocyte transfusion refractoriness due to induced alloimmunization, and increased risk of infection with intracellular pathogens, such as CMV, GTX is not recommended as anti-fungal prophylaxis for all neutropenic patients. However, it may have a role in preventing the progression of existing, severe fungal infections in salvageable patients with anticipated recovery of neutropenia, such as patients post-HCT. Therefore, it is wise and safer, the treating physicians to weigh the risks and benefits and proceed with GTX on a case-by-case basis, at least until large, multicenter randomized clinical trials shed light on the role of GTX as treatment of fungal infections in neutropenic recipients.

#### 3.4.2. Natural Killer and Dendritic Cell Therapy

NK cells, after direct interaction with the pathogen, are activated and exert antifungal activity through the secretion of soluble factors such as perforin and granzyme. Primary NK cells or NK cell lines such as NK-92 exhibited considerable in vitro antifungal activity against medically important fungi, such as *Aspergillus*, *Candida*, mucormycetes, and *Fusarium* [35,36] Recent technical advances allowing for successful expansion of activated NK cells have resulted in reduction of fungal burden in a pulmonary aspergillosis immunocompromised mouse model [37], therefore indicating the translational potential of NK cells as an immunotherapeutic option of IFI, which, however, requires clinical verification.

Dendritic cells (DCs) are a key part of the immune system acting as a bridge between innate and adaptive immunity by capturing pathogens, presenting them to T cells and controlling the polarization of the latter via expression of costimulatory molecules and cytokine secretion. To date, adoptive transfer of *Aspergillus* conidia-pulsed DCs or conidial RNA-transfected DCs induced antifungal Th1 priming and protected post HCT mice from invasive aspergillosis [38]. However, the jury is still out on whether DC vaccination will be effective in patients suffering from IFIs.

## 4. Harnessing T Cells to Control Fungal Infections

T cell immunotherapy has emerged as a promising approach for the treatment of fungal infections in severely immunocompromised patients, such as those undergoing HCT.

In contrast to the early recovery of innate immunity in the first weeks to months after HCT, the reconstitution of adaptive immunity, including fungus-specific T cells (FSTs), can take up to a year [39,40], depriving patients for a long period of time from the natural, infection-fighting army of T cells. Indeed, circulating *Aspergillus*-specific T cells were detected at a low frequency 9 to 12 months after transplantation, displaying a non-protective, low IFN-γ/high interleukin-10 secretion phenotype [41]. Based on the premise that upon encountering a target antigen, pathogen-specific T cells undergo in vivo expansion, the adoptive transfer of FSTs is expected to benefit the immunocompromised patients to rapidly reconstitute their antifungal immunity and effectively tackle emerging fungal threats. The strategy of transferring antigen-specific T cells after allogeneic HCT is well established as prophylaxis and treatment of viral infections; the administration of virus-specific T cells (VSTs), across multiple centers and clinical trials worldwide, has safely and effectively controlled viral infections caused by a broad spectrum of viruses such as cytomegalovirus (CMV), Epstein–Barr virus (EBV), adenovirus, human herpes virus 6 (HHV-6) and polyomavirus type I (BK virus—BKV) or type II (JC virus) [42,43,44,45,46,47,48,49,50,51,52,53,54,55,56,57,58].

Although FSTs have been shown to increase the resistance against *Aspergillus*
*fumigatus* infection or prolong the survival of mice with invasive pulmonary mycosis [59,60,61], as of today, and in contrast to the numerous VST trials, there are only two completed, clinical trials of adoptive T cell immunotherapy against fungal diseases [41,62]. In the first study, conducted 16 years ago, Perrucio et al. showed that the administration of anti-*Aspergillus* T cell clones was safe, enhanced the control of *Aspergillus* antigenemia and improved the survival of transplanted patients with invasive aspergillosis as compared to the control cohort not receiving immunotherapy (survived patients 9/10 vs. 7/13, respectively). Since then, only recently, Gottlieb’s group evaluated the safety and biological efficacy of administering mobilized donor peripheral blood-derived specific T cells with activity against *Aspergillus*, CMV, adenovirus, EBV, varicella-zoster virus (VZV), influenza, and BKV, as prophylaxis of fungal and viral reactivation and/or disease following allogeneic HCT [62]. Grade III/IV graft-versus-host disease (GvHD) was developed in 4/11 patients, however, the early prophylactic administration (median 37 days post-HCT) of multi-pathogen-specific T cells (mp-STs) did not allow the dissociation of the GvHD risk that could be derived from the mp-STs from the GvHD risk originating from the transplant procedure per se, which is expected to be high anyway, early post-transplant. In terms of their efficacy as fungal prophylaxis in particular, none of the enrolled patients including 4 and 2 patients who had positive PCR for *Aspergillus* post HCT and prior to T cell transfer, respectively, required antifungal treatment for invasive fungal infection [62]. An ongoing clinical trial (EudraCT #2013-002914-11) at the Universitätsklinikum Würzburg with highly awaited results, assesses the efficacy of adoptive immunotherapy with donor-derived anti-*Aspergillus* T cells in HCT recipients with probable or proven invasive aspergillosis.

It is thus clear that, despite the early promising clinical results with *Aspergillus*-specific T cells, their use as adoptive immunotherapy for mold infections lags far behind the clinical experience with VST adoptive T cell therapy. The main reasons for the delay in moving this approach to the bedside was the complex, lengthy, labor-intense and costly procedures of manufacturing FSTs [63]. Several groups have now developed good manufacturing practice (GMP)-compliant and less sophisticated methods resulting in significant progress of clinical-grade/scale FST manufacturing [63,64]. Such manufacturing strategies involve T lymphocyte harvest from donor’s blood and subsequently, the isolation of activated T cells after stimulation with the appropriate target-antigens or ex vivo expansion of the endogenous antigen-specific T cells or genetic modification of T cells towards re-directing their specificity via a chimeric antigen receptor (CAR) (Figure 2). Table 1 summarizes the preclinical and clinical studies on the production and administration of FSTs.

### 4.1. Adoptive Immunotherapy with FSTs: The Pros

The recent successes in FST manufacturing paved the way towards clinical application. FST immunotherapy can at least theoretically be superior over the conventional treatment approaches in the battle against fungi (Table 2), for various reasons, mainly associated with the characteristics and potentialities of FSTs.

#### 4.1.1. High Specificity and Potential Efficacy

Unlike the conventional, universal interventions, adoptive immunotherapy involving antigen-specific T cells is a targeted and personalized approach, aiming to eliminate a particular pathogen or multiple pathogens and control the infection on a *per patient* basis. VSTs have documented target-specific, effective control of active viral reactivations and/or diseases post allo-HCT [83,84,85], irrespective of full (graft donor-derived) or partial (third-party donor-derived) HLA-matching between the VST products and patients. In particular, persistent and/or drug-refractory infections/diseases from multiple viruses [86,87,88] and EBV-associated posttransplant lymphoproliferative disorders (PTLD) [89,90], or *de novo* EBV lymphomas [45,91,92,93] were successfully eliminated after VST infusion.

Despite the paucity of clinical data, FSTs, either non-engineered or genetically engineered, have been characterized by strong in vitro specificity against particular fungal antigen(s) (Table 1). Although for the majority of the reported FST products, strong antifungal activity in vitro is provided by the dominant Th1 cytokines IFN-γ or/and TNF-α, other described FSTs additionally produce low to high levels of IL-17, yet an important factor in antifungal immunity [94]. Given that Th17 cells are key players in the host defense against fungal infections, especially at mucosal sites [95,96,97], it remains to be elucidated whether the adoptive transfer of Th17- enriched FSTs will be more powerful against mucosal fungal infections over those with low content in Th17 cells.

Nonetheless, similar to the well-documented efficacy of VSTs and as evidenced by the few published clinical data with FSTs, mold-specific T cells are expected to also have great potential to eradicate mold-infected cells, overcoming the limitations of non-specific pharmacotherapy and the resistance to emerging mold variants.

#### 4.1.2. Broad Protection against Multiple Pathogens

The susceptibility of immunocompromised patients to opportunistic infections by a plethora of pathogens [98], makes them ideal candidates for a treatment providing broad protection against multiple pathogens. The adoptive transfer of multi-targeting antigen-specific T cells may serve as a therapeutic intervention for active infection(s) while providing prophylaxis against other targeted pathogens, not responsible for the infection at the time of cell administration.

The generation of a single T cell product targeting multiple viruses has been pioneered by Leen et al., who initially produced a tri-valent specific product expanded in response to viral challenge and targeting CMV, EBV and adenovirus [51]. This study laid the groundwork for the generation of multi-VSTs with activity extending to VZV [99], BKV, HHV-6 [53] or against respiratory viruses [100,101].

Similarly, the potential of producing multi-FSTs against medically important molds has also been investigated (Table 1). Tramsen et al. generated a memory, Th1 T cell line with tri-fungal specificity against *Candida albicans*, *Aspergillus fumigatus*, and *Rhizopus oryzae* [79]. Likewise, Gottlieb’s group has proved the feasibility of generating multi-FSTs targeting *Aspergillus terreus*, *Candida krusei* and *Rhizopus oryzae* [76] and recently *Candida krusei* and *Aspergillus terreus* [59]. To further unleash the potential of adoptive immunotherapy, Khanna et al. increased the range of targeting against 2–3 viruses and 1–2 fungi [80]. We have also recently shown the generation of GMP-compliant mp-STs simultaneously targeting 3–4 viruses and *Aspergillus fumigatus* [81,82]. These T cell products emerge as a powerful, one-time treatment for severely immunosuppressed patients who suffer from multiple, simultaneous or sequential, life-threatening infections post-transplant.

A key feature of FSTs is their cross-reactivity to different fungi, other than those used for their production. In fact, from the early efforts to develop highly functional Th1 cells against *Aspergillus fumigatus*, Beck et al. advocated that T cells pulsed with a cellular extract derived from *Aspergillus fumigatus* were also responsive against *Aspergillus*
*flavus*, *Aspergillus niger* and *Penicillium chrysogenum* upon cross-challenge [66]. Cross-protective immunity has been previously reported for *Candida* and *Aspergillus fumigatus* [102,103], and has been also observed in the context of pathogen-specific T cells across multiple centers including ours; *Aspergillus* fumigatus-specific T cell products exerted broad anti-fungal cross-immunity to clinical isolates of other *Aspergillus* species (*niger*, *terreus*, and *flavus*), as well as various species of different fungal genera, such as *Candida* (*albicans* and *tropicalis*), *Scedosporium* (*apiospermum* and *prolificans*), *Fusarium* (*solani* and *oxysporum*), *Rhizopus microspores*, *Lichtheimia corymbifera* and *Penicillium* species (Table 1). Ex vivo generated *Rhizopus oryzae*-specific T cells cross-reacted with either *Aspergillus fumigatus* or *Rhizopus (microsporus*, *pusillus)*, *Mucor circinelloides*, *Penicillium chrysogenum* and *Candida albicans* [71,77]. Cross-protective immunity was also documented on multi-FSTs [59,76,79] and mp-STs targeting 3 viruses and *Aspergillus fumigatus* [81,82]. These data suggest an enhanced breadth of fungal recognition, however, whether the broad anti-fungal cross-immunity observed in vitro can provide global fungal protection in vivo, remains to be answered in clinical trials.

#### 4.1.3. Long-Lasting Immunity

Allo-grafted patients are vulnerable to fungal infections even during the late post-transplant period. The existing antifungal armamentarium fails to convey long-term protection leaving those patients at risk for infection recurrence, after cessation of drug therapy [104,105,106]. Longevity is a prominent characteristic of T cell immunity. Following the expansion phase of an immune response, effector CD4+ and CD8+ T cells that survive the contraction phase which usually coincides with the clearance of the antigen, slowly convert into memory cells [107]. This reservoir of cell memory pools is homeostatically maintained and capable of vigorous proliferation following antigen re-encounter, thus acting as a safeguard against possible re-exposure to the same pathogen.

Following the initial report by Riddell et al. [108] revealing the requirement of CD4+ T cells for prolonged survival of adoptively transferred CD8+ T cells in vivo, many groups developed polyclonal CD4+ and CD8+ T cell products. Heslop et al. showed that ex vivo expanded, donor-derived, EBV-specific, polyclonal T cells, expressing an effector memory phenotype, persisted for as long as 12.5 years post-infusion [49] and provided persistent activity against viral reactivation with no evidence of monoclonal outgrowth. Other groups also have demonstrated long-term presence of polyclonal, donor-derived, single- or multi-virus-specific T cells after adoptive transfer [52,109] or stable remission over the years of patients with refractory PTLD receiving EBV-specific T cells [110]. Notably, even in third-party settings, off-the-shelf, partially HLA-matched VSTs persisted up to 12 weeks following infusion, offering durable benefits [86,87], or provided up to 12 months control of previously refractory CMV infections in the majority of treated patients [111]. Given that FSTs share multiple phenotypic and functional characteristics with VSTs in vitro, it could be extrapolated that polyclonal, CD4+ and CD8+ FSTs displaying an effector and central memory phenotype will propagate in vivo upon relevant challenge and successfully eradicate the fungus, promoting a durable and long-lived immunological memory.

#### 4.1.4. Off-the-Shelf Therapy

Recent efforts seem to be geared towards the use of off-the-shelf antigen-specific T cells for adoptive immunotherapy. In this context, antigen-experienced cells have been manufactured from healthy, previously exposed to the targeted antigens individuals, immunophenotypically and functionally characterized, cryopreserved and banked to serve as directly accessible, “on-demand” products for patients who share with the T cell product at least one HLA through which the targeted pathogen is recognized.

Concerning safety, off-the-shelf antigen-specific T cells, have been associated with strikingly low incidences of GvHD and CRS, by virtue of the narrow specificity of endogenous TCRs or attenuated capacity after extended culture to recruit and/or activate macrophages and inflammatory mediators. As regards efficacy, the best possible VST product-recipient matching needs to be considered. In most cases, preference is given to products with the highest HLA matching and mapped antiviral activity through shared HLA antigens.

Infusions of banked, partially HLA-matched VSTs have led to considerable clinical benefit in single or multicenter, phase I and II clinical studies [45,86,87,89,90,93]. There has been no correlation between viral control and the degree of HLA matching (low 1–3 vs. high 4–8 matching alleles) in the HCT context [86,87], even when the treated patient and the VST product were matched only at a single HLA allele, should that allele was mediating strong antiviral activity [86]. In the SOT context, response rates were reported higher as HLA matching was increasing between the donor and the VSTs recipient [90].

The aforementioned studies with off-the-shelf VSTs suggest that multiple infusions may be required due to transient T cell engraftment or no response in some cases; however, a careful selection of T cell products on the basis of partial HLA matching with the recipient and appropriate HLA restriction or switching to another T cell product specific for a different epitope and presented by an alternate shared allele in non-responders, would offer significant clinical benefit against pathogens. Currently, at least two companies, Allovir and Atara Biotherapeutics, are in the process of commercializing such cell products.

Although this strategy has not been tested with FSTs, its adaptation as prophylaxis of high-risk patients or treatment of invasive fungal infections seems promising. Whether however, banked FST cells will have equivalent to VSTs activity against molds, remains to be answered in clinical trials.

#### 4.1.5. Triggering of Epitope Spreading

The risk of pathogen escape from T cell recognition after VST transfer is low [112,113] and further minimized by simultaneous targeting of multiple epitopes and a plethora of antigens for each targeted pathogen. Antigen-experienced T cells offer an added value in the battle against opportunistic infections over other approaches, through the potential induction of what is known as “epitope spreading”. Epitope spreading is a phenomenon whereby a variety of endogenous T cells are activated, diversify and spread their specificity against epitopes other than the ones primarily targeted, either of the same (intramolecular spreading) or a different antigen (intermolecular spreading), thus reinforcing the immune response [114]. Epitope spreading has been detected with both virus-specific and tumor-specific T cells [115]. There are several reports where the infusion of VSTs in patients with malignancies triggered this phenomenon and led to an increase in T cells specific for tumor-associated antigens [115,116] or the infusion of tumor-specific T cells resulted in complete metastatic tumor regression even when the targeted antigen (NY-ESO-1) was not uniformly expressed by the patient’s tumor cells [117].

The potential of epitope spreading as a bystander effect induced by the adoptive transfer of mold-specific T cells, along with their ability to cross-react with a broad spectrum of fungi, may contribute to the development of a compact and powerful T cell product, providing long-term infection control, and the potential to tackle resistant fungal variants.

### 4.2. Adoptive Immunotherapy with FSTs: The Cons

FSTs stand as an attractive strategy against invasive fungal infections, however, certain challenges need to be considered towards moving into the clinic (Table 2).

#### 4.2.1. Selection of the Antigenic Target

The first challenge towards generating non-engineered FSTs is the selection of a proper and GMP-compliant antigenic target. The use of heat-treated conidia or fungal extracts as antigen source seems to be an ideal antigenic stimuli of T cells, as they contain a plethora of antigens and epitopes which in turn can be targeted through a range of HLA types, thus making feasible the generation of FSTs from all donors, irrespective of their HLA alleles. Indeed, heat-treated conidia or lysate of *Aspergillus Fumigatus* were used for the expansion of *Aspergillus*-specific T cells administered in both clinical trials of adoptive immunotherapy with FSTs [41,62]. However, the variability in the quantity of antigens contained in the fungal lysates, along with the potential inclusion of molecules involved in immune evasion [118], makes the preparation of a GMP-compliant fungal lysate complex, demanding and profitless, therefore limiting its commercial availability. GMP-grade overlapping peptide pools from known immunogenic fungal antigens, such as Crf1, Gel1, Pmp20, catalase-1 and SHMT proteins of *Aspergillus* or mp65 for *Candida albicans*, offer an alternative source of antigens for the generation of FSTs for clinical use [72,75,81,82].

#### 4.2.2. Antigenic Competition

Antigenic competition could be a limiting factor in the generation of multivalent antigen-specific T cells. Antigenic competition has been documented in multi-VST products most commonly when derived from CMV-positive donors where there is skewing to a CMV-specific T cell response at the expense of T cell responses of other virus specificities. Such preferential expansion of CMV-specific T cells may be related to the high frequency of CMV-specific T-lymphocytes in peripheral blood of healthy seropositive donors, being several hundred or thousand times higher when compared with circulating T cells of other antigen specificities [119].

Multi-fungus or multi-pathogen-specific T cells may be subjected to antigenic competition especially in the presence of CMV interference. Indeed, Tramsen et al. [79] and Khanna et al. [80] observed a notable decrease in the frequencies of antigen-specific T cells in multi-fungus/pathogen-specific cultures versus single-pathogen cell lines, probably attributed to antigenic competition arising from shared antigen-bearing antigen-presenting cells, as well as from the varying frequencies of subsets of antigen-specific Tcells in the starting PBMCs [120,121]. To overcome antigenic competition, Gottlieb’s group has stimulated individual cultures of viral- and fungal-specific T cells, which they subsequently combined and expanded further after restimulation [62]. We have recently mitigated antigenic competition showing rapid and GMP-compliant generation of a multi-pathogen-specific T cell product as well as a CRISR/CAS9-edited glucocorticoid-resistant multi-pathogen-specific T cell product simultaneously targeting up to 4 viruses and *Aspergillus fumigatus*, to overcome pathogen-specific T cells’ susceptibility to steroids [81,82].

#### 4.2.3. FSTs Outside the Allogeneic Transplantation Context?

While the utility of antigen-specific T cells has been established in the SOT and HCT settings, there is a paucity of data outside the transplant setting. There has been a report on the use of off-the-shelf, EBV-specific T cells in a group of non-transplanted patients with immunosuppression-associated lymphoproliferative diseases who had failed conventional treatments or were too frail to tolerate chemotherapy, showing 64% overall response and 54% survival rates [122]. In addition, partially HLA-matched, BKV–specific T cells administered for progressive multifocal leukoencephalopathy (PML) to a patient with acquired immunodeficiency syndrome resulted in a complete clearance of JC virus in the cerebrospinal fluid and regaining of independent mobility [123]. Although limited, those pilot studies set the scene for considering immunotherapy with FSTs in different settings.

Autologous FST immunotherapy could arise as an alternative to overcome the hurdle of HLA matching. Autologous EBV- and CMV-specific T cells have successfully performed in vivo post SOT [124,125]. However, unlike VSTs, T cells from patients with Paracoccidioidomycosis or Candidemia were shown to express CTLA-4 or other T cell exhaustion markers, respectively [126,127]. Likewise, we observed that autologous T cells derived from patients with invasive aspergillosis, displayed an anergic phenotype that prohibited sufficient ex vivo expansion of *Aspergillus*-specific T cells [78], thus suggesting that adoptive immunotherapy with FSTs may not be feasible in the autologous setting due to failure to proliferate upon antigen encounter and control the infection in vivo.

Another alternative to surmount the constraints of MHC restriction is the use of engineered, mold-specific CAR T cells. A second-generation CAR that incorporates the pattern-recognition ability of Dectin-1 to recognize β-glucans expressed on the cell wall of fungi (D-CAR T cells) was designed to target *Aspergillus fumigatus* and infused D-CAR T cells effectively reduced the fungal burden in an immunocompromised invasive aspergillosis mouse model [73]. Despite the introduced innovation, limitations associated with the potential complications of CAR T cells, should be seriously considered.

#### 4.2.4. “Off-Target” or/and “On-Target Toxicity”

T cell transfer with FSTs derived from the graft source donor and especially a third-party donor, could be associated with an increased risk of alloreactivity against recipient’s cells as a manifestation of “off-target” toxicity, resulting in GvHD. Importantly, GvHD has not been associated with VSTs administration so far, even when used in a third-party setting [128,129,130,131], although it still remains as a possibility, should tissue upregulation of HLA class II molecules in an inflammatory environment occurs, facilitating the presentation of cross-reacting antigens to FSTs [132].

On the other hand, in the presence of high pathogen burden, there is a risk of “on-target” toxicity attributed to an excessive immune response against the targeted pathogens, such as CRS or immune reconstitution inflammatory syndrome (IRIS). CRS is a systemic and potentially life-threatening inflammatory response frequently observed with CAR T cell administration and its severity usually correlates with the anti-target potency of the treatment [133]. IRIS occurs during the course of various invasive fungal diseases in immunodeficient hosts at the time of reversal of immune deficiency, is triggered by the recovery of immune cells and presented as CRS and excessive host inflammatory response [134].

CRS is only rarely observed when unmodified antigen-specific T cells are administered [135,136], thus it is not anticipated to arise using non-engineered anti-fungal T cells recognizing targets through their endogenous T cell receptor. On the contrary, in the case of D-CAR T cells targeting fungi, CRS will be rather expected as an “on target” toxicity, manageable although with IL6 blockade (tocilizumab). Nonetheless, even with the unmodified anti-fungal T cells, neither CRS nor IRIS, could be ruled out.

Additionally, the otherwise advantage of FSTs to cross-react with other than the targeted fungi thus broadening their targeting repertoire involves the “off-target” risk of their activation by harmless commensal or symbionts, such as gut mycobiota. This activation could potentially lead to ‘dysbiosis’ and an imbalance of immune homeostasis, which may subsequently promote inflammation or drive pathogenesis [137,138,139]. Although it has not yet been addressed if such CD4+ T cell activation is directly driving pathogenesis or it is simply an epiphenomenon to preceding intestinal barrier disruption and a similar “off-target” effect has not been reported after administration of Aspergillus-specific T cells [41,62], activation of FSTs toward mycobiota cannot be excluded.

#### 4.2.5. Sustainability of the Clinical Benefit?

Another concern when using a “living drug” as a treatment strategy is to ensure that this “drug” survives and remains functional in the recipient’s, often hostile, microenvironment. Since immunosuppressive compounds, such as cyclosporine, mycophenolic acid and steroids, which are routinely used in SOT or HCT patients, impair optimal T cell functionality [140,141,142,143,144], we have previously suggested that adoptive T cell therapy in the transplantation context could exhibit its full potential when anti-GvHD prophylaxis is not required, such as in cases of T cell depleted HCT [63]. To allow transplanted patients with opportunistic viral infections to enjoy the benefits of adoptive immunotherapy, regardless of the intensity of immunosuppression, groups have recently genetically modified VSTs to render them resistant to glucocorticoids [145,146,147]. In this setting, we edited the glucocorticoid receptor (GR) and developed a powerful T cell product, called “Cerberus” T cells, exhibiting multiple specificities against 4 viruses (adenovirus, CMV, EBV and BKV) and *Aspergillus Fumigatus*, along with resistance to glucocorticoids [82]. The genetic modification, was highly on-target with very low or meaningless off-target cutting while the generated GR-resistant mp-STs shared similar phenotypic and functional characteristics to their unedited counterparts. In the unlikely case of induced alloreactivity, the retained susceptibility to complement-mediated lysis by other immunosuppressants (anti-thymocyte globulin), allows their elimination on demand [82]. Even under intense immunosuppression, “Cerberus” T cells targeted 5 pathogens and also cross-reacted with other *Aspergillus* genera (flavus and niger) and fungi species (*Fusarium; Oxysporum* and *Solani*), suggesting strong potential if used as prophylaxis or treatment of fungal infections in transplanted and highly immunocompromised patients. Finally, while the use of banked T cell products constitutes a safe and attractive plan for the management of opportunistic infections and diseases, its efficacy is limited by a shorter, approximately 12 weeks, in vivo persistence as compared to graft donor-derived cells. A strategy to overcome the inappropriate alloreactive T cell response eventually leading to rejection of partially-HLA matched T cell products has recently been proposed by Mo et al. They engineered a chimeric 4-1BB-specific alloimmune defense receptor (ADR) that enabled ADR-expressing T cells to selectively target activated T and NK cells eliminating alloreactive lymphocytes but spare their resting counterparts evading immune rejection [148]. This approach for reduced recognition of therapeutic cells by the host’s cellular immunity could be adapted for the generation of FSTs to overcome the limitations of short-term persistence or repeated infusions of banked FSTs.

## 5. Conclusions

There is an increasing prevalence of IFIs in immunodeficient or transplanted patients who are highly prone to fungal infections. As the outcomes of IFIs in these patients are poor, the transfer and subsequent reconstitution of antifungal immunity might enable overcoming the underlying deficiency causing the infection rather than simply suppress the fungal growth by pharmacological agents. Advances in adoptive immunotherapy with VSTs including simpler and rapid manufacturing, underscore a potential therapeutic opportunity for FSTs in the prevention and treatment of IFIs. The results of the only two reported up-to-date clinical studies with FSTs have been very promising, however additional clinical trials are needed to provide solid evidence of mold-specific T cell safety and efficacy. The availability of GMP-compliant and simple protocols for the generation of clinical doses of T cells recognizing a single mold or a wide spectrum of viral and fungal targets, their potentiality for broad anti-fungal coverage via cross-immunity, the successful use of banked antigen-specific T cells on a best HLA-matched basis to treat viral infections, and the recent development of tools to further reinforce FSTs by rendering them resistant to immunosuppressive agents or fortifying them to avoid rejection, suggest that adoptive immunotherapy with FSTs has a major role to play in improving the often dismal outcomes of patients with IFIs.

## Figures and Tables

**Figure 1 jof-07-00451-f001:**
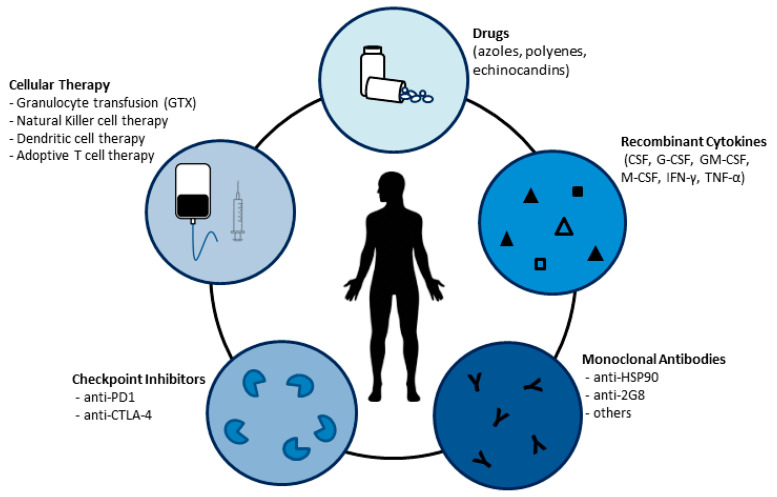
Standard and alternative therapeutic approaches for the management of invasive fungal infections.

**Figure 2 jof-07-00451-f002:**
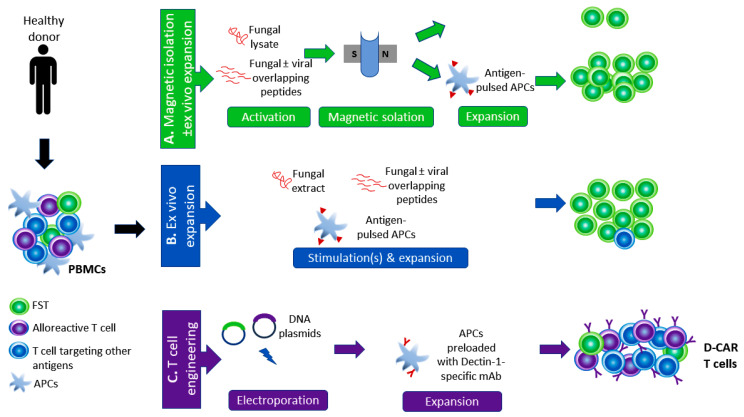
Strategies for mono-fungus-, multi-fungus- or multi-pathogen-specific T cell manufacturing. A: PBMCs collected from a healthy donor can be activated with fungal extracts or with overlapping peptide libraries of fungal and viral antigens and directly be selected ex vivo with monoclonal antibody capture of cytokine-producing cells or antibodies binding to activation antigens attached to magnetic beads. Selected cells can subsequently be expanded in culture. B: Alternatively, FSTs or multi-pathogen-specific T cells can be expanded ex vivo upon stimulation with fungal extracts or with overlapping peptide libraries of fungal and viral antigens. C: Finally, FSTs can be produced by genetically modifying T cells to express a chimeric antigen receptor (CAR) that redirects their specificity to a desired antigen, e.g., Dectin-1 (D-CAR T cells). FST: fungal-specific T cells; PBMCs: peripheral blood mononuclear cells; APCs: antigen-presenting cells; mAb: monoclonal antibody.

**Table 1 jof-07-00451-t001:** Preclinical and clinical studies of the generation of fungus-specific T cells.

	Targeted Pathogen(s)	Manufacturing Protocol	Group	Antigen Competition	Cross-Reactivity	Clinically Tested
Monofungus-specific T cell products	*A. fumigatus*	Ex vivo expansion post stimulation of PBMCs with *A. fumigatus* extract	Ramadan et al., 2004 [65]	n/a	Not tested	No
	*A. fumigatus*	Ex vivo expansion post stimulation of PBMCs with *A. fumigatus* extract	Perrucio et al., 2005 [41]	n/a	Not tested	Yes
	*A. fumigatus*	IFN-γ selection post stimulation of PBMCs with *Aspergillus* extract	Beck et al., 2006 [66]	n/a	Against *A. flavus*, *A. niger*, and *P. chrysogenum*-not against *A. alternata* and *C. albicans*	No
	*C. albicans*	IFN-γ selection and ex vivo expansion post stimulation of PBMCs with *C. albicans* extract	Tramsen et al., 2007 [67]	n/a	Against *C. tropicalis* but not *C. glabrata*, *P. chrysogenum*, *A. alternata*, and *A. fumigatus*	No
	*A. fumigatus*	Ex vivo expansion post stimulation of PBMCs with *A. fumigatus* Af16	Zhu et al., 2008 [68]	n/a	Not tested	No
	*A. fumigatus*	IFN-γ selection and ex vivo expansion post stimulation of WBC with *A. fumigatus* extract	Tramsen et al., 2009 [69]	n/a	Limited cross-reactivity against other filamentous fungi (data not shown), but not *Candida* species	No
	*A. fumigatus*	Ex vivo expansion post stimulation of PBMCs with *A. fumigatus* lysate	Gaundar et al., 2012 [70]	n/a	Against *A. niger* and *A. flavus*, *C. albicans*, *S. apiospermum* and *Penicillium* species—not against *A. terreus*, *C. glabrata*, *Fusarium species* and *Mucor* species	No
	*R. oryzae*	CD154 selection, ex vivo expansion and IFN-γ enrichment post stimulation of PBMCs with *R. oryzae* cell extract	Schmidt et al., 2012 [71]	n/a	Against *R. microsporus*, *R. pusillus*, *M. circinelloides*, *A. fumigatus*, *P. chrysogenum* and *C. albicans.* No response against *A. flavus*, *A. alternata* and *M. racemosus*	No
	*A. fumigatus*	CD137 selection and ex vivo expansion post stimulation of PBMCs with Crf1 and catalase-1 (*A. fumigatus*)	Jolink et al., 2013 [72]	n/a	Not tested	No
	*Aspergillus species*	Dectin-1 CAR T cells specific for β-glucans and bispecific T cells co-expressing Dectin-1-CAR and CD19-CAR	Kumaresan et al., 2014 [73]	n/a	Not tested	No
	*A. fumigatus*	CD137 selection post stimulation of PBMCs with *A. fumigatus* lysate	Bacher et al., 2015 [74]	n/a	Against other *Aspergillus* spp. and *C. albicans*	No
	*A. fumigatus*	CD154 or CD137 selection and ex vivo expansion post stimulation of PBMCs with fungal extract or Crf1 and/or Pmp20 and/or Gel1	Stuehler et al., 2015 [75]	n/a	Against *A. flavus*, *A. terreus*, and *A. niger*, *S. apiospermum*, *S. prolificans*, *F. solani*, *R. microsporus*, *L. corymbifera*, and *C. albicans*	No
	*A. fumigatus* or *A. flavus* or *A. terreus* or *C. albicans* or *C. krusei* or *F. solani* or *F. oxysporum* or *R. oryzae* or *L. prolificans*	Εx vivo expansion post stimulation of PBMCs with fungal extract	Deo et al., 2016 [76]	n/a	*A terreus*, *F oxysporum* and *L prolificans* T cell cultures cross-reacted with one another and against *A. fumigatus*, *A. flavus* and *F. solani. C. krusei* T cell cultures cross-reacted against *C. albicans*, or with *Aspergillus* and *Fusarium* species. Cross-reactivity with *R. oryzae* was observed in a subset of the T cell cultures	No
	*R. oryzae*	Ex vivo expansion post stimulation of PBMCs with *R. oryzae* lysate	Castillo et al., 2018 [77]	n/a	Against *A. fumigatus*	No
	*A. fumigatus*	Ex vivo expansion post stimulation of PBMCs with *A. fumigatus* lysate or Crf1, Gel1 and SHMT	Papadopoulou et al., 2019 [78]	n/a	Against *A. flavus*, *A. niger*, *F. solani*, *F. oxysporum*, *C. tropicalis* and *C. albicans*	No
Multifungus specific T cell products	*A. fumigatus*, *C. albicans* and *R. oryzae*	IFN-γ selection and ex vivo expansion post stimulation of PBMCs with fungal extract	Tramsen et al., 2013 [79]	Lower numbers of T cells responding to *A. fumigatus* were detected in the multifungus T cell product	Against *A. niger*, *P. chrysogenum*, *C. tropicalis*, *M. circinelloides*, *R. pusillus*, *R. microsporus* and *R. microsporus-oligosporus*	No
	*A. terreus*, *C. krusei* and *R. oryzae*	Εx vivo expansion ±TNF-α selection post stimulation of PBMCs with fungal extract	Deo et al., 2016 [76]	Not illustrated	Against *A. fumigatus*, *C. albicans*, *C. krusei* and *Lomentospora*, *A. flavus* and *R. oryzae*	No
	*C. krusei* and *A. terreus*	CD137 selection and ex vivo expansion post stimulation of PBMCs with fungal extract	Castellano-Gonzalez et al., 2020 [59]	Not illustrated	Higher cross-reactivity against *A. fumigatus*, *A. flavus*, *A. terreus*, *C. albicans* and *C. krusei* and lower against *F. solani*, *F. oxysporum* and *S. prolificans*	No
Multi-pathogen-specific T cell products	AdV, EBV, CMV, *C. albicans*, and/or *A. fumigatus* or AdV, EBV, *C. albicans*, and *A. fumigatus*	CD154 selection and ex vivo expansion post stimulation of PBMCs with hexon (AdV), LMP2 (EBV), pp65 (CMV), mannose protein 65 (MP65, *C. albicans*) and Crf1 (*A. fumigatus*)	Khanna et al., 2011 [80]	Notable decrease in the frequencies of antigen-specific T cells in the multipathogen-specific cultures vs. single lines	Not tested	No
	EBV, CMV, BKV, and *A. fumigatus*	Ex vivo expansion post stimulation of PBMCs with IE1 and pp65 (CMV), EBNA1, LMP2 and BZLF1 (EBV), Large T and VP1 (BKV) and Crf1, Gel1 and SHMT (*A. fumigatus*)	Papadopoulou et al., 2021 [81]	No	Cross-reactivity against *C. albicans*, *C. tropicalis*, *F. solani*, *F. oxysporum*	No
	AdV, EBV, CMV, BKV, VzV, influenza and *A. fumigatus*	Ex vivo expansion post stimulation of PBMCs with pp65 (CMV), Hexon (AdV), EBNA1, LMP2A and BZLF1 (EBV), Large T and VP1 (BKV), Vzv vaccine, Influenza vaccine and lysate (*A. fumigatus*). Cultures from individually stimulated products were combined and restimulated for further expansion.	Gottllieb et al., 2021 [62]	n/a	Not tested	Yes
	AdV, EBV, CMV, BKV, and *A. fumigatus*—steroid resistant	Ex vivo expansion post stimulation of PBMCs with hexon and penton (AdV), IE1 and pp65 (CMV), EBNA1, LMP2 and BZLF1 (EBV), Large T and VP1 (BKV) and Crf1, Gel1 and SHMT (*A. fumigatus*) and genetical modification to inactivate the glucocorticoid receptor	Koukoulias et al., 2021 [82]	No	Cross-reactivity against *A. flavus*, *A. niger*, *F. solani*, *F. oxysporum*	No

AdV: adenovirus; *A. alternata*: *Alternaria alternata*; *A. fumigatus*: *Aspergillus fumigatus*; *A. flavus*: *Aspergillus flavus*; *A. niger*: *Aspergillus niger*; *A. terreus*: *Aspergillus terreus*; BKV: BK virus; *C. albicans*: *Candida albicans*; *C. glabrata*: *Candida glabrata*; *C. krusei*: *Candida krusei*; *C. tropicalis*: *Candida tropicalis*; CAR: Chimeric antigen receptor; CMV: cytomegalovirus; EBV: Epstein Barr virus; *F. solani*: *Fusarium solani*; *F. oxysporum*: *Fusarium oxysporum*; IFN-γ: Interferon-γ; *L. corymbifera*: *Lichtheimia corymbifera*; *L. prolificans*: *Lomentospora prolificans*; *M. circinelloides*: *Mucor circinelloides*; *M. racemosus*: *Mucor racemosus*; *P. chrysogenum*: *Penicillium chrysogenum*; PBMCs: peripheral blood mononuclear cells; *R. microsporus*: *Rhizopus microsporus*; *R. oryzae*: *Rhizopus oryzae*; *R. pusillus*: *Rhizomucor pusillus*; *S. apiospermum*: *Scedosporium apiospermum*; *S. prolificans*: *Scedosporium prolificans*; TNF-α: tumor necrosis factor-α; Vzv: Varicella zoster.

**Table 2 jof-07-00451-t002:** Advantages and considerations of adoptive immunotherapy with FSTs.

Pros	Cons
High specificity and potential efficacy	Selection of the appropriate antigenic target
Broad coverage against multiple pathogens	Antigenic competition
Long-lasting immunity with graft donor-derived FSTs	“Off-target” or/and “on-target” toxicity
Immediate availability as an off-the-shelf therapy	No long-term persistence of anti-fungus immunity with third-party, off-the shelf FSTs
Triggering of epitope spreading	Sustainability of the clinical benefit?

## Data Availability

Not applicable.

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
