# Peer review of "Reinforcing the Immunocompromised Host Defense against Fungi: Progress beyond the Current State of the Art"

_jof, 2021, doi:10.3390/jof7060451_

Round 1

Reviewer 1 Report

The manuscript by Karavalakis and colleagues provides information regarding the use of adjunctive therapies, particularly immune-based therapies in the treatment of life-threatening fungal infections in the immunocompromised host. The focus is on T cell-based therapies. Overall, the authors provide some new information, but they fail to adequately discuss the downsides of this approach. The comparison to viral infections in immunocompromised hosts has some validity but the fungi are more complex and therefore offer more chances of these types of therapies being deleterious.

  1. They fail to mention adequately the problems with developing antigen-specific antifungal T cells. First, one needs a defined antigenic stimulus preferably a cocktail of antigens that can be manufactured reproducibly and will pass GMP. One cannot merely make extracts from fungi and hope they contain the right quantity of antigens and no inhibitory substances. This process has to be standardized. The authors need to mention this.
  2. The authors do talk somewhat about off-target effects but inadequately. For example, since many fungi reside in the gut, it is entirely possible that some of these T cells will hone to the gut recognize the fungus or another species of the same fungus and attack and enhance inflammation in the gut. This scenario is much less likely with viruses such as CMV or EBV than with fungi. The same is true for using CART cells with Dectin-1. Gut attack may limit applicability.
  3. The authors focus on Th1 but Th17 are critically important for mucosal pathogens. Therefore, one would like to be able to transfer in Th17 cells and hope they maintain phenotype. The authors should discuss when Th1 and when Th17 should be deployed.

There are a few minor concerns.

  1. The English should be improved.
  2. The authors state that IFN-gamma polarizes naïve CD4+ cells to Th1 antifungal cells. That statement is incorrect. IL-12 is critically important for polarization and IFN would not necessarily polarize to Th1 antifungal cells. There is nothing specific about polarization in terms of antigen unless the driving force for polarization is an antigen.
  3. 5FC is thrown into the mix and added to the polyene section giving the false impression that it may be a polyene which of course it is not. Beyond that, 5FC does have anti-Candida activity just that one would never use it alone.
  4. The authors in their introductory paragraph discuss life-threatening IFIs and refer to genera. While there is a Pneumocystis genus the only species that attacks humans is jirovecii. That is not a genus. Cryptococcus, Candida, Aspergillus yes those are genera and many species infect humans. But not Pneumocystis.

Reviewer 2 Report

Review for article entitled: Reinforcing the immunocompromised host defense against fungi: state of the art and beyond.

The review focuses on treatment of fungal infections. The authors discuss the conventional methods and alternative therapies such as T cell immunotherapies. 

Overall opinion:

To me it seems as though the review is unnecessarily lengthy and a bit repetitive. Some sections could be cropped: For example the overview of the antifungal drugs - some of the information is inaccurate and confusing (see specific comments). It may be better to either crop this section focusing only on the biologics or substantially improve it.  Sections 7.2 and 7.3 are highly repetitive and could be re-written to be one section. Section 8.4 also seems a little repetitive.

Lines of the manuscript were not numbered so I will refer to sections.

Specific Comments:

  1. Page 2: section 2.1 lanosterol 14 alpha demethylase belongs to the CYP450 superfamily of enzymes rather than being a CYP450-dependent
  2. Page 2: section 2.1  Inhibition of lanosterol demethylase by azoles does not lead to cell lysis. Azoles have more of a fungistatic effect, preventing fungal cell replication and altering the composition of the cell membrane due to lack of ergosterol. 
  3. Page 2: section 2.1 In fact the whole paragraph is poorly written, another example is “ Because of the cytochrome p450 inhibition, they interfere with the metabolism of other drugs but they are overall well tolerated.” - should clarify that this is referring to the human liver enzymes. In addition Last line of this paragraph - posaconazole, it is fine to mention that it's effective against mucormycosis but it is not characterized by that, in fact posaconazole is frequently used as a prophylactic treatment in at-risk patients. 
  4. Page 3: Section 3.1 - is it worth having a section with 3 lines? This could be part of the previous or the subsequent paragraphs.
  5. Page 3, section 3.2 - the final sentence in this section does not make any sense, please revise.
  6. Page 4: Section 3.4 “It was tested to treat candidemia in combination with amphotericin B with promising results in Candida-associated mortality, but with serious safety and quality issues.” - are the serious safety issues stemming from Amphotericin B or the antibody? And what are the quality issues? The last sentence in this paragraph - ‘eliminate current antifungal strategies drawbacks’- could the authors expand on this statement?
  7. Section 4.1 Do the authors foresee any side effects/ complications with GTX?
  8. Section 4.2. Was the NK cell therapy tested on mice or was this only tested in vitro? 
  9. Section 7.2 Would there be any problems associated with GR deletion? Not stressing this point.
  10. A figure outlining the process for generating FSTs or VSTs would be helpful.
  11. Perhaps a table with the adoptive therapy pros and cons will be beneficial? 

Minor comments:

  1. References 2 and 5 are the same publication
  2. Section 7 should read: Adoptive immunotherapy with FSTs: the pros. 
  3. There are a number of misspelt words etc but because the lines aren’t numbered it's too tedious to bring up all.

Reviewer 3 Report

This is a review article based on the expertise of the senior author`s group on advanced therapies devoted to overcome invasive aspergillosis in immunocompromised patients. This is a challenging aim that is correctly addressed along the text. I only have a few comments that mainly aim to minor points. Some aspects of English writing should be corrected.

  1. Extend the section dealing with IFNϒ as an adjunctive therapy in invasive mycosis, since it seems play some role in patients with STAT3 deficiency that compromise IFNϒ and Th17 responses (PMID: 32047500).
  2. There are some inconsistencies for the use of italic types when referring to genus and species.
  3. The reference to laminarin as a component of the for the β-glucan layer of Candida cell-wall is inappropriate (3.4. Section). Laminarin refers to the β-glucan-based soluble polysaccharide present in some algae. It is currently used to inhibit the binding of yeast β-glucan to its cell receptors.
  4. The description of Dectin-1 in 8.2 targeting β-glucans should be reworded since it does not clearly depict that Dectin-1 is a β-glucan receptor.

Round 2

Reviewer 1 Report

The authors have addressed my concerns.